

# Elevated serum FGF21 levels predict heart failure during hospitalization of STEMI patients after emergency percutaneous coronary intervention

Lingyun Gu, Wenxi Jiang, Wenlong Jiang, Zhuowen Xu, Weizhang Li and Hua Zhang

The Jiangyin Clinical College, Xuzhou Medical University, Jiangyin, Jiangsu, China

Corresponding authors
Weizhang Li,
liweizhang1131@163.com
Hua Zhang, ryzhanghua@163.com

## ABSTRACT

**Background:** Fibroblast growth factor 21 (FGF21) has multiple cardioprotective effects including modulation of glucolipid metabolism, anti-inflammation, and anti-oxidative stress, but its association with the heart failure during hospitalization in patients with ST-segment elevation myocardial infarction (STEMI) undergoing emergency percutaneous coronary intervention (PCI) has not been reported.
**Methods:** A total of 348 STEMI patients treated with emergency PCI were included from January 2016 to December 2018. Relevant biochemical indicators were measured by central laboratory. Serum FGF21 levels were measured by ELISA. The occurrence of heart failure during hospitalization was recorded. Patients' cardiac function was assessed by echocardiography.
**Results:** Serum FGF21 levels were significantly higher in the STEMI group with heart failure than in the group without heart failure ($249.95 \pm 25.52$ *vs.* $209.98 \pm 36.35$, $P < 0.001$). Serum FGF21 levels showed a strong positive correlation with N-terminal precursor B-type natriuretic peptide (NT-proBNP) in STEMI patients ($r = 0.749$, $P < 0.001$). FGF21 was found to be an independent risk factor for the development of heart failure during hospitalization in STEMI patients by binary logistic regression analysis. The area under curve (AUC) for FGF21 to predict the development of heart failure during hospitalization in STEMI patients was 0.816 (95% CI [0.770–0.863]) according to the receiver operating characteristic (ROC) curve analysis.
**Conclusion:** Elevated serum FGF21 levels have been shown to be a strong predictor of heart failure during hospitalization in patients with STEMI after emergency PCI.

## INTRODUCTION

Although the widespread clinical application of early cardiac reperfusion therapy has significantly improved the in-hospital mortality and clinical prognosis of acute myocardial infarction (AMI), AMI is still one of the most common causes of heart failure during hospitalization due to myocardial cell necrosis, mechanical complications, and ischemia-reperfusion injury, with a prevalence of 14% to 36% (*Bahit, Kochar & Granger, 2018*; *Jenca et al., 2021*). N-terminal proB-type natriuretic peptide (NT-proBNP) was

found to be significantly associated with heart failure during hospitalization after AMI, but it was susceptible to age, gender, abdominal obesity, and glomerular filtration rate (*Mayr et al., 2011*; *Qin et al., 2021*). Therefore, it is imperative to actively search for biomarkers that are closely associated with AMI and that can predict heart failure during hospitalization.

AMI is a common cause of heart failure (*Bahit, Kochar & Granger, 2018*). After AMI, myocardial cells experience impaired energy metabolism, inflammatory response, oxidative stress, ischemia-reperfusion injury, and myocardial hypertrophy and fibrosis leading to adverse myocardial remodeling and promoting the development of heart failure (*Berezin & Berezin, 2020*; *Frantz et al., 2022*). It was found that the heart not only secretes fibroblast growth factor 21 (FGF21), but is also one of its target organs (*Planavila et al., 2013b*). In acute myocardial ischemia, cardiomyocytes can either secrete FGF21 in the form of paracrine secretion or induce FGF21 secretion from liver and adipose tissue, which acts on cardiomyocytes in an endocrine manner to exert cardioprotective effects (*Chiba et al., 2018*; *Liu et al., 2013*; *Liu et al., 2012*). FGF21 not only inhibits the formation of vascular atherosclerotic plaques, but also plays a cardioprotective role by reducing myocardial ischemia-reperfusion injury and improving post-infarction cardiac function through anti-inflammation, anti-oxidative stress, and regulation of energy metabolism (*Hu et al., 2018*; *Tanajak, Chattipakorn & Chattipakorn, 2015*). It was found that in the rat model of AMI, the antioxidant capacity of rats with high expression of FGF21 was significantly enhanced, and the inflammatory response and fibrosis of the heart were inhibited, while this effect was significantly weakened after knocking out FGF21 (*Li et al., 2021*; *Ma et al., 2021*).

Serum FGF21 levels were significantly elevated in patients with AMI, coronary artery disease, hypertension, and dilated cardiomyopathy (*Gu et al., 2021a*, *2021b*; *Zhang et al., 2021*). Moreover, serum FGF21 levels were also significantly elevated in patients with Heart failure with preserved ejection fraction (HFpEF) and end-stage heart failure (*Di Lisa & Itoh, 2015*; *Ianos et al., 2021*; *Planavila et al., 2015*). Furthermore, serum FGF21 levels were associated with reinfarction within 30 days and long-term MACE (*Chen, Lu & Zheng, 2018*; *Sunaga et al., 2019*; *Zhang et al., 2015*). However, there are no studies correlating serum FGF21 levels with heart failure after AMI during hospitalization in ST segment elevation myocardial infarction (STEMI) patients treated with emergency percutaneous coronary intervention (PCI). Therefore, this study intends to investigate the correlation between serum FGF21 levels and heart failure during hospitalization in STEMI patients treated with emergency PCI by measuring serum FGF21 levels.

## METHODS

### Study population

This study is a secondary analysis of a previously published prospective study. Unlike previously published studies focusing on the long-term prognosis of FGF21 in STEMI patients, this study focused on the relationship between FGF21 and heart failure during hospitalization in STEMI patients. The study was approved by the Ethics Committee of

Jiangyin People's Hospital (approval number: 2015ER035), and written informed consent was obtained from all enrolled patients prior to participation.

We first retrospectively analyzed STEMI patients from January to December 2015 and found that the incidence of heart failure during hospitalization in STEMI patients was approximately 30%. Then we performed sample size calculation by PASS 15.0 software. We chose the confidence interval model for one proportion for the calculation, setting the α value of 0.05 and also setting the β value of 0.1 with two-sided, which yielded a sample size of 341 at a $P$ value of 0.3. This means that a sample size of 341 produces a two-sided 95% confidence interval with a width equal to 0.1 when the sample proportion is 0.3.

Inclusion criteria for this study: (1) STEMI patients within 12 h of onset; (2) meeting the ACC/AHA guidelines for emergency PCI. STEMI was diagnosed based on typical chest pain symptoms combined with dynamic evolution of ECG and cardiac enzymes. Exclusion criteria were cardiac shock, primary valvular heart disease, congenital heart disease, tachycardia-induced cardiomyopathy, pericardial disease, chronic liver insufficiency, acute renal failure, rheumatic disease, pulmonary embolism, neoplasm, inflammatory or infectious disorders, and excess alcohol consumption. Finally, 348 patients with STEMI were actually enrolled from January 2016 to December 2018. The patients enrolled in this study were the same as those studied in previous published articles by our group (*Gu et al., 2021a*). All STEMI patients underwent emergency PCI within 12 h of the onset of ischemic symptoms and were treated medically according to the American College of Cardiology/ American Heart Association guidelines (*O'Gara et al., 2013*).

The mean hospitalization duration for STEMI patients in this study was 11.77 days. According to the 2013 ACCF/AHA guidelines for the management of heart failure, combining patient symptoms, signs, NT-proBNP, and cardiac ultrasound. The diagnostic process of heart failure during hospitalization in patients with STEMI is detailed in Fig. 1. First, STEMI patients present with symptoms of chest tightness and shortness of breath, combined with signs of insufficient cardiac output and stasis in the physical or pulmonary circulation. Second, patients' NT-proBNP levels were tested and heart failure was determined by stratifying the patients according to their age. Finally, the patient's cardiac ultrasound was combined to determine the presence of myocardial systolic dysfunction and whether LVEF was decreased. Judgment of whether heart failure occurs in patients with STEMI is made by the above means.

## Clinical and laboratory assessments

An experienced physician took the medical history of the enrolled patients. A portion of the elbow venous blood drawn in the early morning of day 2 on an empty stomach was sent to the central laboratory for troponin I levels by Beckman DXI800 and for triglyceride, total cholesterol, low density lipoprotein cholesterol (LDL-C), high density lipoprotein cholesterol (HDL-C), uric acid, NT-proBNP and creatinine levels by Roche e602 and c701 modules, respectively. Another part of the blood sample was obtained as serum and stored at −80 °C. Serum FGF21 concentrations were measured using a quantitative human ELISA kit (DF2100; R&D Systems, Minneapolis, MN, USA) as in our previously published article (*Gu et al., 2021b*).

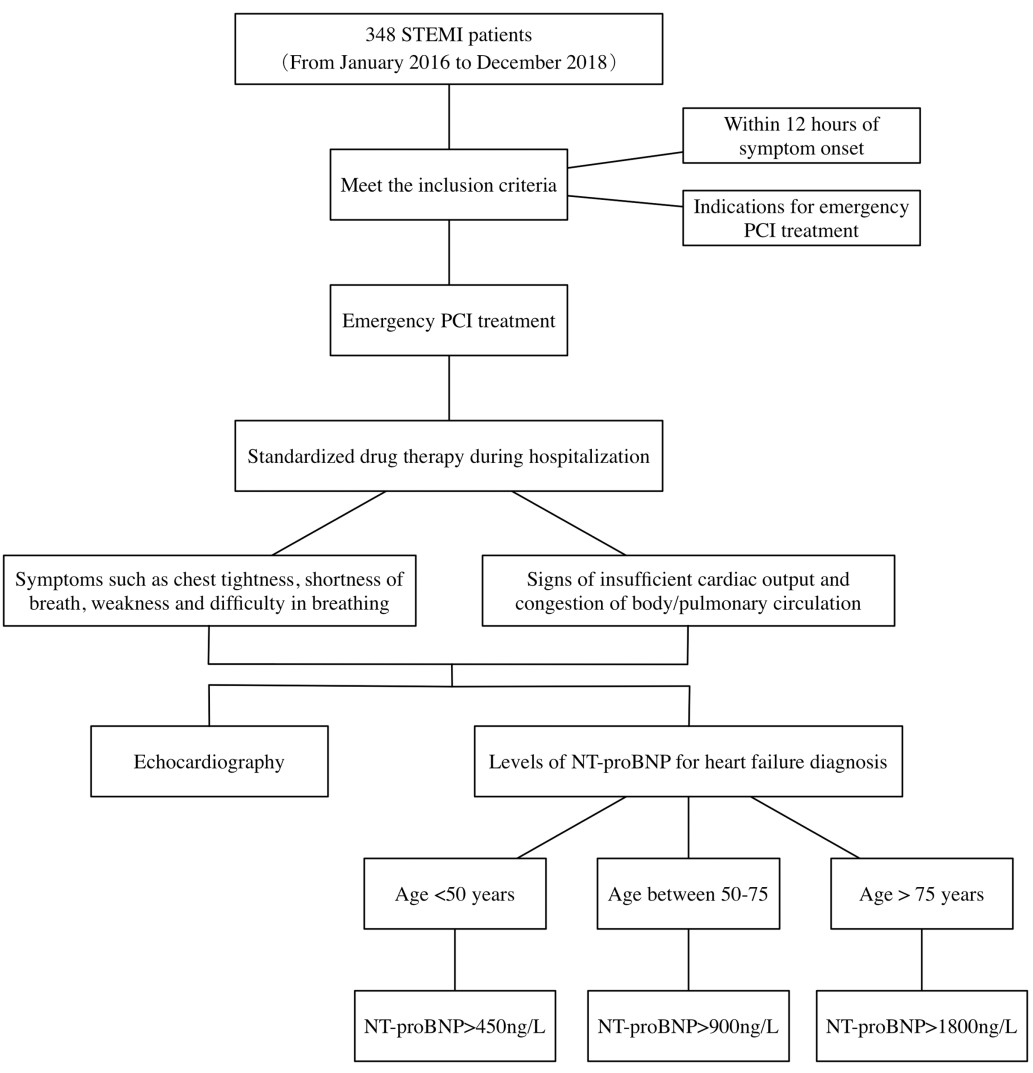

**Figure 1 Diagnosis process of heart failure in STEMI patients during hospitalization.** STEMI, ST segment elevation myocardial infarction; PCI, percutaneous coronary intervention; NT-proBNP, N-terminal proB-type natriuretic peptide.

## Echocardiography

Echocardiography (Philips iE 33 xMatrix) was performed on all STEMI patients 7–10 days after emergency PCI. According to the American Society of Echocardiography guidelines, the method of flow convergence (PISA) assesses the severity of mitral regurgitation (*Zoghbi et al., 2003*). As in our previous study, pulmonary artery pressure, left atrial dimension (LAD), interventricular septum thickness (IVST), left ventricular posterior wall thickness (LVPWT), left ventricular end-diastolic diameter (LVEDD), left ventricular systolic diameter (LVESD), left ventricular ejection fraction (LVEF), left ventricular end systolic volume (LVESV), and left ventricular end diastolic volume (LVEDV) were measured by echocardiography. The left ventricular mass (LVM) was calculated by the following formula: 0.8 × 1.04 × [(IVST + LVPWT + LVEDD) 3 − (LVEDD) 3] + 0.6 (*Lang et al., 2015*).

## Statistical analysis

Sample size calculation was performed by PASS15.0 software. Statistical analysis was performed by SPSS 22.0 statistical package. Quantitative variables were presented as mean ± standard deviation and then Student's t test was performed for comparison. Categorical variables were expressed as absolute numbers (percentages) and then compared by chi-square test. The correlation between FGF21 and clinical variables was determined by Spearman correlation analysis. For STEMI patients who developed heart failure, we first performed a univariate regression analysis to identify statistically significant indicators. Follow-up regression analysis of these indicators as covariates for multiple factors was performed to explore the predictors of heart failure occurrence during hospitalization in STEMI patients. The predictive value of FGF21 for the development of heart failure in STEMI patients was analyzed by Receiver operating characteristic (ROC) curves. The area under curve (AUC) of NT-proBNP and FGF21 were compared by Z-test with MedCalc 20.00. $P$ value < 0.05 was considered statistically significant (two-tailed).

## RESULTS

A total of 131 STEMI patients developed heart failure during hospitalization among 348 STEMI patients. The comparison of clinical information between STEMI with and without heart failure is summarized in Table 1.

The LDL-C, troponin I, NT-proBNP, FGF21 levels, LAD, LVEDD, LVESD, and LVESV were higher in the heart failure group than in the no heart failure group, while the HDL-C levels and LVEF were lower than in the no heart failure group (Table 1) ($P$ < 0.05). There was no statistical difference between the two groups in terms of sex, age, previous hypertension, diabetes mellitus, history of atrial fibrillation, blood total cholesterol, triglyceride, uric acid, creatinine level, pulmonary artery pressure, mitral regurgitation, IVST, LVPWT, LVEDV.

Spearman analysis showed that serum FGF21 levels were correlated with age, LDL-C, HDL-C, troponin I, NT-proBNP levels, LVEDD, LVESD, LVESV, and LVEF. Among them, FGF21 was negatively correlated with HDL-C and LVEF, and the rest variables were positively correlated. Notably, FGF21 showed a strong positive correlation with NT-proBNP (r = 0.749, $P$ < 0.001) (Table 2). Serum FGF21 levels were not significantly correlated with gender, previous hypertension, diabetes mellitus, history of AF, total blood cholesterol, triglycerides, uric acid, creatinine levels, pulmonary artery pressure, mitral regurgitation, LAD, IVST, LVPWT, LVEDV, LVM, and coronary multiple lesions.

In the univariate logistic regression analysis, the occurrence of heart failure in STEMI patients was used as the dependent variable, and gender, age, presence of hypertension, diabetes mellitus, atrial fibrillation, blood cholesterol, triglycerides, LDL-C, HDL-C, uric acid, creatinine, troponin I, NT-proBNP, FGF21, pulmonary pressure, mitral regurgitation, LAD, IVST, LVPWT, LVEDD, LVESD, LVEDV, LVESV, LVM, LVEF, and multiple coronary lesions as independent variables. The univariate Logistic regression analysis found that LDL-C, HDL-C, troponin I, NT-proBNP, FGF21, LAD, LVEDD, LVESD, LVESV, LVM, LVEF were associated with heart failure during hospitalization in STEMI patients ($P$ < 0.05) (Table 3).

**Table 1 Comparison of clinical data between the heart failure group and the no heart failure group.**

| Variables | No HF group (n = 217) | HF group (n = 131) | P value |
|---|---|---|---|
| **Demographic data** | | | |
| Male, n (%) | 174 (80.18%) | 106 (80.92%) | 0.868 |
| Age (years) | 61.06 ± 12.36 | 63.70 ± 13.99 | 0.075 |
| Hypertension, n (%) | 112 (51.61%) | 73 (55.73%) | 0.456 |
| Diabetes mellitus, n (%) | 35 (16.13%) | 29 (22.14%) | 0.161 |
| Atrial fibrillation, n (%) | 20 (9.22%) | 7 (5.34%) | 0.191 |
| **Laboratory data** | | | |
| Total cholesterol (mmol/L) | 4.34 ± 1.20 | 4.29 ± 0.95 | 0.682 |
| Triglyceride (mmol/L) | 1.85 ± 1.59 | 1.78 ± 1.15 | 0.656 |
| LDL-C (mmol/L) | 3.09 ± 0.93 | 3.55 ± 01.07 | 0.000 |
| HDL-C (mmol/L) | 1.19 ± 0.46 | 1.06 ± 0.40 | 0.007 |
| Uric acid (μmol/L) | 344.16 ± 95.20 | 333.55 ± 102.80 | 0.329 |
| Creatinine (μmol/L) | 76.39 ± 21.18 | 88.34 ± 93.60 | 0.152 |
| Troponin I (ng/ml) | 38.87 ± 19.65 | 52.69 ± 19.71 | 0.000 |
| NT-proBNP (pg/ml) | 336.10 ± 352.25 | 1,019.00 ± 641.83 | 0.000 |
| FGF21 (pg/ml) | 209.98 ± 36.35 | 249.95 ± 25.52 | 0.000 |
| **Echocardiographic data** | | | |
| Pulmonary pressure (mmHg) | 32.01 ± 7.97 | 32.80 ± 8.94 | 0.407 |
| Mitral regurgitation | 32 (15.31%) | 17 (13.82%) | 0.712 |
| LAD (mm) | 40.71 ± 4.32 | 42.20 ± 5.28 | 0.008 |
| IVST (mm) | 9.75 ± 1.52 | 9.75 ± 1.96 | 0.994 |
| LVPWT (mm) | 9.78 ± 1.28 | 10.01 ± 1.45 | 0.128 |
| LVEDD (mm) | 51.98 ± 4.00 | 53.19 ± 5.05 | 0.024 |
| LVESD (mm) | 36.69 ± 4.23 | 39.54 ± 5.26 | 0.000 |
| LVEDV (ml) | 125.64 ± 33.24 | 129.73 ± 44.88 | 0.367 |
| LVESV (ml) | 56.01 ± 19.32 | 65.64 ± 28.26 | 0.001 |
| LVM (g) | 189.56 ± 40.42 | 199.44 ± 45.25 | 0.041 |
| LVEF (%) | 55.75 ± 7.08 | 50.39 ± 7.12 | 0.000 |
| **Multiple coronary lesions, n (%)** | 96 (44.24%) | 45 (34.35%) | 0.069 |

Note:

STEMI, ST segment elevation myocardial infarction; FGF21, fibroblast growth factor 21; HF, heart failure; LDL-C, low density lipoprotein cholesterol; HDL-C, high density lipoprotein cholesterol; NT-proBNP, N-terminal proB-type natriuretic peptide; LAD, left atrial dimension; IVST, interventricular septal wall thickness; LVPWT, left ventricular posterior wall thickness; LVEDD, left ventricular end-diastolic diameter; LVESD, left ventricular systolic diameter; LVEDV, left ventricular end-diastolic volume; LVESV, left ventricular end-systolic volume; LVM, left ventricular mass; LVEF, left ventricular ejection fraction.

In the multivariate logistic regression analysis, the occurrence of heart failure in STEMI patients was used as the dependent variable, and the parameters that were statistically significant in the univariate logistic regression analysis (LDL-C, HDL-C, troponin I, NT-proBNP, FGF21, LAD, LVEDD, LVESD, LVESV, LVM, and LVEF) as independent variables, and the Forward: LR approach was selected for logistic stepwise regression analysis. The multivariate logistic regression analysis found that the elevated levels of

**Table 2  Correlation analysis of FGF21 with clinical variables.**

| Variables | r$_s$ | P value |
|---|---|---|
| **Demographic data** | | |
| Male, *n* (%) | −0.022 | 0.681 |
| Age (years) | 0.235 | 0.000 |
| Hypertension, *n* (%) | 0.071 | 0.187 |
| Diabetes mellitus, *n* (%) | 0.098 | 0.066 |
| Atrial fibrillation, *n* (%) | 0.017 | 0.758 |
| **Laboratory data** | | |
| Total cholesterol (mmol/L) | −0.078 | 0.148 |
| Triglyceride (mmol/L) | 0.017 | 0.747 |
| LDL-C (mmol/L) | 0.220 | 0.000 |
| HDL-C (mmol/L) | −0.145 | 0.007 |
| Uric acid (μmol/L) | −0.038 | 0.481 |
| Creatinine (μmol/L) | 0.075 | 0.160 |
| Troponin I (ng/ml) | 0.357 | 0.000 |
| NT-proBNP (pg/ml) | 0.749 | 0.000 |
| **Echocardiographic data** | | |
| Pulmonary pressure (mmHg) | 0.008 | 0.885 |
| Mitral regurgitation | 0.017 | 0.760 |
| LAD (mm) | 0.080 | 0.885 |
| IVST (mm) | −0.107 | 0.051 |
| LVPWT (mm) | −0.064 | 0.245 |
| LVEDD (mm) | 0.118 | 0.031 |
| LVESD (mm) | 0.137 | 0.012 |
| LVEDV (ml) | 0.096 | 0.074 |
| LVESV (ml) | 0.113 | 0.035 |
| LVM (g) | −0.017 | 0.751 |
| LVEF (%) | −0.127 | 0.021 |
| **Multiple coronary lesions, *n* (%)** | −0.022 | 0.685 |

Note:
STEMI, ST segment elevation myocardial infarction; FGF21, fibroblast growth factor 21; LDL-C, low density lipoprotein cholesterol; HDL-C, high density lipoprotein cholesterol; NT-proBNP, N-terminal proB-type natriuretic peptide; LAD, left atrial dimension; IVST, interventricular septal wall thickness; LVPWT, left ventricular posterior wall thickness; LVEDD, left ventricular end-diastolic diameter; LVESD, left ventricular systolic diameter; LVEDV, left ventricular end-diastolic volume; LVESV, left ventricular end-systolic volume; LVM, left ventricular mass; LVEF, left ventricular ejection fraction.

NT-proBNP and FGF21 and decreased levels of LVEF may be independent risk factors for heart failure during hospitalization in STEMI patients ($P < 0.05$) (Table 4).

We selected FGF21 and NT-proBNP, a classical marker of heart failure, for ROC curve analysis. According to the ROC curve analysis, the AUC of NT-proBNP to predict the occurrence of heart failure during hospitalization in STEMI patients was 0.929 (95% CI [0.901–0.957]) (Fig. 2), while the AUC of FGF21 was 0.816 (95% CI [0.770–0.863]). The AUCs of logNT-proBNP and logFGF21 were statistically significant (Z = 5.293, $P < 0.001$). The FGF21 has a similar role to NT-proBNP in predicting the development of

**Table 3 Univariate logistic analysis of heart failure in STEMI patients.**

| Variables | β | W | P value | OR | 95% CI |
|---|---|---|---|---|---|
| **Demographic data** | | | | | |
| Male | 0.047 | 0.028 | 0.868 | 1.048 | [0.605–1.814] |
| Age | 0.016 | 3.348 | 0.067 | 1.016 | [0.999–1.033] |
| Hypertension | 0.165 | 0.554 | 0.457 | 1.180 | [0.763–1.824] |
| Diabetes mellitus | 0.391 | 1.951 | 0.162 | 1.478 | [0.854–2.559] |
| Atrial fibrillation | −0.587 | 1.672 | 0.196 | 0.556 | [0.228–1.353] |
| **Laboratory data** | | | | | |
| Total cholesterol | −0.041 | 0.169 | 0.681 | 0.960 | [0.788–1.168] |
| Triglyceride | −0.035 | 0.199 | 0.655 | 0.965 | [0.827–1.127] |
| LDL-C | 0.461 | 16.230 | 0.000 | 1.585 | [1.267–1.983] |
| HDL-C | −0.814 | 6.963 | 0.008 | 0.443 | [0.242–0.811] |
| Uric acid | −0.001 | 0.954 | 0.329 | 0.999 | [0.997–1.001] |
| Creatinine | 0.004 | 1.993 | 0.158 | 1.004 | [0.998–1.011] |
| Troponin I | 0.035 | 33.657 | 0.000 | 1.035 | [1.023–1.048] |
| NT-proBNP | 0.004 | 72.197 | 0.000 | 1.004 | [1.003–1.005] |
| FGF21 | 0.039 | 67.061 | 0.000 | 1.040 | [1.031–1.050] |
| **Echocardiographic data** | | | | | |
| Pulmonary pressure | 0.011 | 0.689 | 0.407 | 1.011 | [0.985–1.038] |
| Mitral regurgitation | −0.120 | 0.137 | 0.712 | 0.887 | [0.470–1.675] |
| LAD | 0.067 | 7.401 | 0.007 | 1.070 | [1.019–1.123] |
| IVST | 0.001 | 0.000 | 0.994 | 1.001 | [0.877–1.142] |
| LVPWT | 0.127 | 2.290 | 0.130 | 1.136 | [0.963–1.340] |
| LVEDD | 0.063 | 5.576 | 0.018 | 1.065 | [1.011–1.122] |
| LVESD | 0.131 | 23.829 | 0.000 | 1.140 | [1.082–1.202] |
| LVEDV | 0.003 | 0.943 | 0.332 | 1.003 | [0.997–1.009] |
| LVESV | 0.019 | 12.728 | 0.000 | 1.109 | [1.008–1.029] |
| LVM | 0.005 | 4.117 | 0.042 | 1.005 | [1.000–1.011] |
| LVEF | −0.108 | 33.862 | 0.000 | 0.898 | [0.865–0.931] |
| **Multiple coronary lesions** | −0.416 | 3.298 | 0.069 | 0.660 | [0.421–1.034] |

Note:

STEMI, ST segment elevation myocardial infarction; FGF21, fibroblast growth factor 21; LDL-C, low density lipoprotein cholesterol; HDL-C, high density lipoprotein cholesterol; NT-proBNP, N-terminal proB-type natriuretic peptide; LAD, left atrial dimension; IVST, interventricular septal wall thickness; LVPWT, left ventricular posterior wall thickness; LVEDD, left ventricular end-diastolic diameter; LVESD, left ventricular systolic diameter; LVEDV, left ventricular end-diastolic volume; LVESV, left ventricular end-systolic volume; LVM, left ventricular mass; LVEF, left ventricular ejection fraction.

heart failure during hospitalization in STEMI patients. The optimal cut-off value of FGF21 was calculated as 234.55 pg/ml based on the Jorden index, with a sensitivity of 87.8% and a specificity of 70%.

## DISCUSSION

In this study, we observed that serum FGF21 levels were associated with NT-proBNP in STEMI patients after emergency PCI and were elevated in STEMI patients who developed heart failure during hospitalization. Elevated serum FGF21 levels are an independent risk

**Table 4 Multivariate logistic analysis of heart failure in STEMI patients.**

| Variables | β | W | P value | OR | 95% CI |
|---|---|---|---|---|---|
| NT-proBNP | 0.003 | 31.965 | 0.000 | 1.003 | [1.002–1.005] |
| FGF21 | 0.019 | 9.326 | 0.002 | 1.019 | [1.007–1.032] |
| LVEF | −0.147 | 30.375 | 0.000 | 0.863 | [0.819–0.910] |

Note:
STEMI, ST segment elevation myocardial infarction; CI, confidence interval; FGF21, fibroblast growth factor 21; NT-proBNP, N-terminal proB-type natriuretic peptide; LVEF, left ventricular ejection fraction.

factor for the development of heart failure during hospitalization in STEMI patients and have predictive value.

It was found that the cardioprotective effect of FGF21 was limited due to its low expression in the normal heart (*Chen et al., 2021*). Once the heart is exposed to multiple stimuli such as ischemia and hypoxia, inflammation, oxidative stress, lipotoxicity and endoplasmic reticulum stress, the secretion of FGF21 is significantly increased to fully play a cardioprotective role (*Planavila et al., 2013a*; *Planavila, Redondo-Angulo & Villarroya, 2015*). However, FGF21, which has cardioprotective effects, predicts the incidence and prognosis of coronary heart disease and AMI (*Gu et al., 2021a*; *Lakhani et al., 2018*). It was found that the reason for this paradoxical phenomenon may be that elevated FGF21 in pathological states may be a compensatory response to underlying stress, or it may be caused by FGF21 resistance due to impaired FGF21 signaling (*Lewis et al., 2019*; *Woo et al., 2013*). This cardioprotective and prognostic prediction of cardiovascular disease by FGF21 is similar to that of BNP.

While BNP and NT-proBNP are recommended by the American and European Heart Associations as biomarkers for the diagnosis and prediction of heart failure, recombinant BNP is used in the treatment of heart failure because of its cardioprotective effects (*Heidenreich et al., 2022*; *McDonagh et al., 2021*). Consistent with previous studies, the present study observed that NT-proBNP was an independent risk factor for the development of heart failure during hospitalization in patients with STEMI and had a predictive value for the development of heart failure. Although FGF21 is not a better predictor of heart failure during hospitalization than NT-proBNP in patients with STEMI after emergency PCI, this study is the first to find that FGF21 is similar to NT-proBNP in predicting heart failure during hospitalization and may be used as a complement to NT-proBNP for the prediction of heart failure.

Cardiac myocytes in STEMI patients treated by emergency PCI are subject to impaired energy metabolism, inflammatory response, oxidative stress, and ischemia-reperfusion injury that cause myocyte necrosis, myocardial stenosis, arrhythmias, and mechanical complications that induce or exacerbate the development of heart failure (*Berezin & Berezin, 2020*; *Buja, 2005*). FGF21 was found to upregulate glucose transporter-1 (GLUT1) expression during acute myocardial ischemia, thereby improving energy supply to cardiac myocytes, increasing cell migration and decreasing apoptosis (*Hu, Cao & Liu, 2017*). After myocardial infarction, inflammatory cytokines show aggregation in the area near the infarct, with increased expression of pro-inflammatory factors and decreased expression of

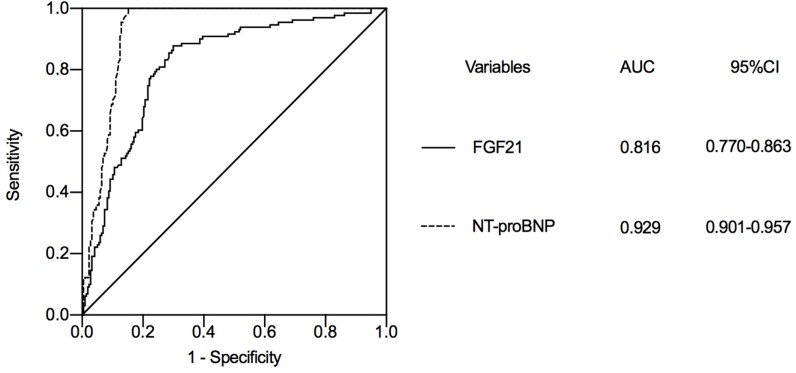

**Figure 2 The value of FGF21 and NT-proBNP in predicting heart failure during hospitalization in STEMI patients after emergency PCI.** STEMI, ST segment elevation myocardial infarction; PCI, percutaneous coronary intervention; FGF21, fibroblast growth factor 21; NT-proBNP, N-terminal proB-type natriuretic peptide; AUC, area under curve; CI, confidence interval.

anti-inflammatory mediators, while exogenous FGF21 can effectively inhibit the inflammatory response and fibrosis in the post-infarction heart, thus exerting cardioprotection (*Li et al., 2021*). In addition, FGF21 protected cardiomyocytes from I/R injury through miR-145 and autophagy, reduced morphological changes in cardiomyocytes, regulated apoptosis and cell migration, and inhibited inflammatory responses, which significantly reduced infarct size and improved cardiac function (*Hu et al., 2018*). Thus, FGF21 can exert cardioprotective effects through multiple mechanisms, thereby improving structural remodeling and reducing the occurrence of heart failure, and is promising as a potential cytokine for the treatment of heart failure (*Redondo-Angulo et al., 2017*).

The present study has some limitations. First, this study is a single-center study. The collection of cases was performed at only one hospital, which may have a degree of selection bias. Second, no follow-up was performed in this study. Whether FGF21 can predict the long-term prognosis of STEMI patients after emergency PCI needs to be further investigated, and this is the focus of our follow-up study.

## CONCLUSIONS

In this study, we observed significantly elevated serum FGF21, NT-proBNP and troponin I levels in STEMI patients after emergency PCI who developed heart failure. However, after logistic regression analysis, FGF21 was found to have a similar effect to NT-proBNP in predicting the development of heart failure during hospitalization in STEMI patients. Therefore, FGF21 may be a potential biomarker for the development of heart failure during hospitalization in STEMI patients after emergency PCI, and it has predictive value for the development of heart failure.

### Funding
The research was supported by development fund of Affiliated Hospital of Xuzhou Medical University (XYFY2021029). The funders had no role in study design, data collection and analysis, decision to publish, or preparation of the manuscript.

### Grant Disclosures
The following grant information was disclosed by the authors:
Affiliated Hospital of Xuzhou Medical University: XYFY2021029.

### Competing Interests
The authors declare that they have no competing interests.

### Author Contributions

- Lingyun Gu conceived and designed the experiments, performed the experiments, prepared figures and/or tables, and approved the final draft.
- Wenxi Jiang performed the experiments, prepared figures and/or tables, and approved the final draft.
- Wenlong Jiang analyzed the data, prepared figures and/or tables, and approved the final draft.
- Zhuowen Xu analyzed the data, authored or reviewed drafts of the article, and approved the final draft.
- Weizhang Li conceived and designed the experiments, authored or reviewed drafts of the article, and approved the final draft.
- Hua Zhang conceived and designed the experiments, performed the experiments, authored or reviewed drafts of the article, and approved the final draft.

### Human Ethics
The following information was supplied relating to ethical approvals (*i.e.*, approving body and any reference numbers):
The ethics committee of Jiangyin People's Hospital approved this study (2015ER035).

### Data Availability
The raw data is available as a Supplemental File.

### Supplemental Information
Supplemental information for this article can be found online at http://dx.doi.org/10.7717/peerj.14855#supplemental-information.

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
