# Peer review of "Elevated serum FGF21 levels predict heart failure during hospitalization of STEMI patients after emergency percutaneous coronary intervention"

_PeerJ, doi:10.7717/peerj.14855_

## Round 0.1 · original submission · Major Revisions

· Academic Editor

Major Revisions

Please address the reviewers' comments as noted below. Thank you.

Reviewer 1 ·

Basic reporting

No comment

Experimental design

No comment

Validity of the findings

I do note a statistical difference in the levels of troponin elevation in the patients with heart failure group. FGF21 has been shown to have a protective role in the myocardial ischemia-reperfusion injury. A note of this finding should be included in the conclusion since patients with a larger area of myocardial infarction may have elevated FGF21 levels and may confound the study findings.

Reviewer 2 ·

Basic reporting

Good basic reporting with professional english, aims , references, structure and data.

Experimental design

1.Relatively well defined, but still unclear whether the authors recommend FGF-21 as surrogate for heart failure and elevated NT-pro BNP or if there are other added benefits to checking the marker.

2. Like already pointed out by the authors, will need follow up data to see whether there is any prediction of outcomes in the future.

Validity of the findings

1.The article talks about heart failure and elevated FGF-21, while the authors are trying to establish that elevated FGF-21 correlates with heart failure, but there is no mention of correlation with extent and severity of acute MI and if there is a correlation with troponin elevation.

2. Though the authors point out similar elevation of FGF-21 to NT-proBNP but are not able to point out any differences between these two lab tests in terms of outcomes and severity.

3. The authors mention correlation with NT-proBNP and may be complementary, but if it doesn't any thing more than severity of reduced EF and how does it complement NT-proBNP

·

Basic reporting

No comment

Experimental design

No comment

Validity of the findings

No comment

Additional comments

Overall paper looks good, well designed, with appropriate stats, however there are some changes that I would recommend.

Minor changes only:

Abstract--> you start with the word although which means that majority of the physicians are already aware of this fact, which might not be true. You should also expand on cardioprotective effects as permitted in an abstract.
You could start as “ FGF21 has multiple cardioprotective effects like ………, but its association with heart failure…… has not been reported”

Title--> Expand the word STEMI as ST elevation myocardial infarction the 1st time you use it in abstract or introduction. What are the chances that FGF21 elevation in MI are just an association with heart failure ? do you have data to prove it otherwise

Line 43--> can use the word imperative instead of important.

From Line 46--> can you please expand about the effects of FGF21 on the heart and myocardium, can expand the physiological and pathological effects/benefits of FGF21 levels. Can make use of these papers.

Ma Y, Kuang Y, Bo W, Liang Q, Zhu W, Cai M, Tian Z. Exercise Training Alleviates Cardiac Fibrosis through Increasing Fibroblast Growth Factor 21 and Regulating TGF-β1-Smad2/3-MMP2/9 Signaling in Mice with Myocardial Infarction. Int J Mol Sci. 2021 Nov 15;22(22):12341. doi: 10.3390/ijms222212341. PMID: 34830222; PMCID: PMC8623999.

Li J, Gong L, Zhang R, Li S, Yu H, Liu Y, Xue Y, Huang D, Xu N, Wang Y, Xu Y, Zhao Y, Li Q, Li M, Li P, Liu M, Zhang Z, Li X, Du W, Wang N. Fibroblast growth factor 21 inhibited inflammation and fibrosis after myocardial infarction via EGR1. Eur J Pharmacol. 2021 Nov 5;910:174470. doi: 10.1016/j.ejphar.2021.174470. Epub 2021 Aug 31. PMID: 34478691.

Tanajak P, Chattipakorn SC, Chattipakorn N. Effects of fibroblast growth factor 21 on the heart. J Endocrinol. 2015 Nov;227(2):R13-30. doi: 10.1530/JOE-15-0289. Epub 2015 Sep 4. PMID: 26341481.


Line 54--> What are the variety of CVD Can you please expand on these.

Can you also please tell how Heart failure occurs in Acute MI, and how FGF-1 is protective.


You were expecting 600 patients, but only 348 were included. Did you check FGF-21 on all continuous patients, or were any of the patients excluded from checking FGF21 ?


Line 92, Line 93--> sentence seems incomplete and doesn’t make sense. Please expand on the diagnosis of heart failure !

Line 100--> ‘occurred’ instead of occurs. This whole statement should not come at the end, but somewhere earlier in the paragraph. Please blend this paragraph in a better way. Thanks.

Line 180 --> ‘correlated’ instead of associated.

Line 196- line 205 --> Please explain- In what conditions are FGF21 better than proBNP and in what conditions are probnp better than FGF21. Why should doctors start using FGF21 ? please expand.

Line 230 --> whats the best time for FGF21 check. What day post AMI ?

Is FGF21 vs proBNP cost effacious in terms of cost of lab check !

In the table- 2nd box--> Is it supposed to be meet or ‘met the inclusion criteria’

I hope FGF21 would be used more often and more studies are done to replicate in various ethnicities across the globe.

---

## Round 0.2 · Minor Revisions

· Academic Editor

Minor Revisions

As this is a retrospective study, please explain how the consent was obtained from each patient?

·

Basic reporting

Much improved abstract, introduction of FGF21 in correlation to AMI and heart failure. all revisions were appropriate

Experimental design

Good.

Validity of the findings

good. no changes suggested.

Additional comments

Article looks much better after revisions. No changes suggested.

---

## Round 0.3 · Minor Revisions

· Academic Editor

Minor Revisions

Thank you for responding to the query. In that case, we recommend to say that the study is a part of prospective study in the methods section. Saying that this study is retrospective adds to confusion. Please clarify in the first paragraph of the methods that this is a secondary analysis of a prior published prospective study as the consent was not obtained solely for this particular study

---

## Round 0.4 · accepted · Accept

· Academic Editor

Accept

Congratulations. Your article is now accepted for publication.